# Single molecule delivery into living cells

Chalmers C. Chau[1,2,3], Christopher M. Maffeo [4,5], Aleksei Aksimentiev [4,5], Sheena E. Radford [1], Eric W. Hewitt [1] ✉ & Paolo Actis [2,3] ✉

Controlled manipulation of cultured cells by delivery of exogenous macromolecules is a cornerstone of experimental biology. Here we describe a platform that uses nanopipettes to deliver defined numbers of macromolecules into cultured cell lines and primary cells at single molecule resolution. In the nanoinjection platform, the nanopipette is used as both a scanning ion conductance microscope (SICM) probe and an injection probe. The SICM is used to position the nanopipette above the cell surface before the nanopipette is inserted into the cell into a defined location and to a predefined depth. We demonstrate that the nanoinjection platform enables the quantitative delivery of DNA, globular proteins, and protein fibrils into cells with single molecule resolution and that delivery results in a phenotypic change in the cell that depends on the identity of the molecules introduced. Using experiments and computational modeling, we also show that macromolecular crowding in the cell increases the signal-to-noise ratio for the detection of translocation events, thus the cell itself enhances the detection of the molecules delivered.

The cell, the fundamental unit of life, is a microscopic chemical reactor in which thousands of processes happen simultaneously to bring about biological function, all within a highly crowded and compartmentalised environment[1]. The manipulation of cells enabled by the controlled delivery of biological macromolecules is an indispensable tool for studying these cellular processes[2,3]. An array of methods has been developed to deliver molecules into mammalian cells, including encapsulation into lipid vesicles and viral-like particles, chemical transfection, electroporation, and microinjection[2–4].

Microinjection uses glass micropipettes with micron-size tips to mechanically penetrate the plasma membrane, enabling molecules to be delivered[2–4]. However, the large size of the micropipette's tip (typically ranging from 0.5 to 75 μm in external diameter)[5] relative to the cell can cause significant perturbation during injection, disrupting the actin cytoskeleton and deforming the morphology of the cell[6]. One solution is to reduce the probe size, and several approaches have been developed that use nanoscale probes, including fluidic force microscopy (FluidFM)[7,8], nanostraws[9–11], carbon-nanotube cellular endoscopes[6,12], nanofountain probe electroporation[13,14], nanopore electroporation[15,16], electrowetting injection[17,18], nanoneedles[19–21], polymeric nanoneedles[22], and electroactive

nanotubes[23,24]. Herein, we use nanopipettes as a nanoscale injection probe. Nanopipettes can be easily fabricated at the bench with tuneable pore diameter down to 10 nm (a nanopore) and by fitting nanopipettes with electrodes, the translocation of molecules into the cell is controlled by the application of a suitable voltage. For this, the polarity and magnitude of the voltage will depend on the charge of the molecule being delivered, with the magnitude not exceeding ±1 V[25–27]. Moreover, the small size of the nanopipette's tip size relative to the cell means that cell survival is much improved compared to microinjected cells[28–31].

The translocation of a single macromolecule through the pore at the tip of a nanopipette results in a detectable alteration in the measured ionic current, which can be used to characterise and quantify the number of molecules that pass through the nanopore[32–39]. In recent work, we have shown that the detection of nucleic acids and proteins is enhanced by their translocation into a polymer–electrolyte bath containing poly(ethylene) glycol (PEG)[33,36], and have shown that this arises from a combination of unique ion transport behaviour and the interaction between the translocating molecule and the polymer–electrolyte interface[38,39]. Inspired by these observations, we here describe the development of a nanoinjection platform in which

[1]School of Molecular and Cellular Biology and Astbury Centre for Structural Molecular Biology, University of Leeds, Leeds LS2 9JT, UK. [2]School of Electronic and Electrical Engineering and Pollard Institute, University of Leeds, Leeds LS2 9JT, UK. [3]Bragg Centre for Materials Research, University of Leeds, Leeds, UK. [4]Department of Physics, University of Illinois at Urbana–Champaign, Urbana, IL 61801, USA. [5]Beckman Institute for Advanced Science and Technology, University of Illinois at Urbana-Champaign, Urbana, IL 61801, USA. ✉e-mail: e.w.hewitt@leeds.ac.uk; p.actis@leeds.ac.uk

nanopipettes are used to perform the quantitative delivery of biological macromolecules into the highly crowded interior of mammalian cells that induce different cellular responses.

In our nanoinjection platform, a nanopipette is integrated into a scanning ion conductance microscope (SICM), where it functions both as a scanning probe and an injection probe[40–43]. The SICM enables the automated positioning of the nanopipette's tip on the surface of a living cell with nanometre resolution[42]. The nanopipette can then be inserted into either the nucleus or cytoplasm and the delivery of molecules into the cells is then triggered by the application of an appropriate voltage. We demonstrate that the nanoinjection platform can be used for both cell lines and primary cells to perform quantitative delivery of DNA, globular proteins, and protein fibrils, all at single molecule resolution and that the macromolecules retain function after delivery into the cells. Furthermore, we show that the ionic current signatures are enhanced for single molecule delivery into the macromolecular crowded environment of either the cell interior or an electrolyte solution with a high concentration of bovine serum albumin (BSA). By using computational modelling, we demonstrate that this enhancement is caused by an increased concentration of analyte at the nanopipette opening after translocation and the displacement of crowding molecules near the nanopore.

## Results

### Overview of the nanoinjection platform

The nanoinjection platform comprises five components: (1) a SICM, (2) a nanopipette which acts as the SICM scanning probe, a nanoinjection tool and as a single molecule counter (Supplementary Fig. 1), (3) microstepper motors for coarse positioning of the nanopipette in the region of interest (4) piezoelectric actuator for positioning the nanopipette in three dimensions with nanoscale precision, and (5) a spinning disk confocal microscope for brightfield and fluorescence imaging before and after nanoinjection (Fig. 1A).

In the nanoinjection procedure, brightfield and fluorescence microscopy are used to identify the cell for injection, and the microstepper motor performs the initial coarse positioning of the nanopipette at the site of injection. Fine positioning of the nanopipette uses the SICM control software and piezoelectric actuators (Fig. 1B). SICM relies on the measurement of the ionic current between an Ag/AgCl electrode inserted in the nanopipette, and a reference electrode immersed in an electrolytic solution where the sample is placed[40–42]. By applying a voltage between the two electrodes, an ionic current flows through the nanopore at the tip of the nanopipette. When the nanopipette approaches a surface, the measured ionic current drops. This current drop is proportional to the separation between the nanopore and the sample and can be used as active feedback to maintain the nanopipette-sample distance constant (Supplementary Fig. 2)[42]. A topographic image of the cell surface can be obtained, both pre- and post-injection, by recording the vertical position of the probe over each scanned pixel[42]. Integration of the nanopipette with piezoelectric actuators allows the positioning of the probe in the three dimensions with nanoscale precision. Thus, by determining the height of the plasma membrane by SICM, the nanopipette can then be inserted into the cell to a pre-determined depth (Fig. 1B). Once inserted, the application of an appropriate voltage drives the movement of molecules from inside the nanopipette into the cell. By measuring the disruption in the ionic current flow caused when individual molecules of sufficient size pass through the pore at the nanopore's tip, the number of molecules delivered into the cell can be quantified (Fig. 1B). The nanopipette is then retracted from the cell and, as appropriate, the cell

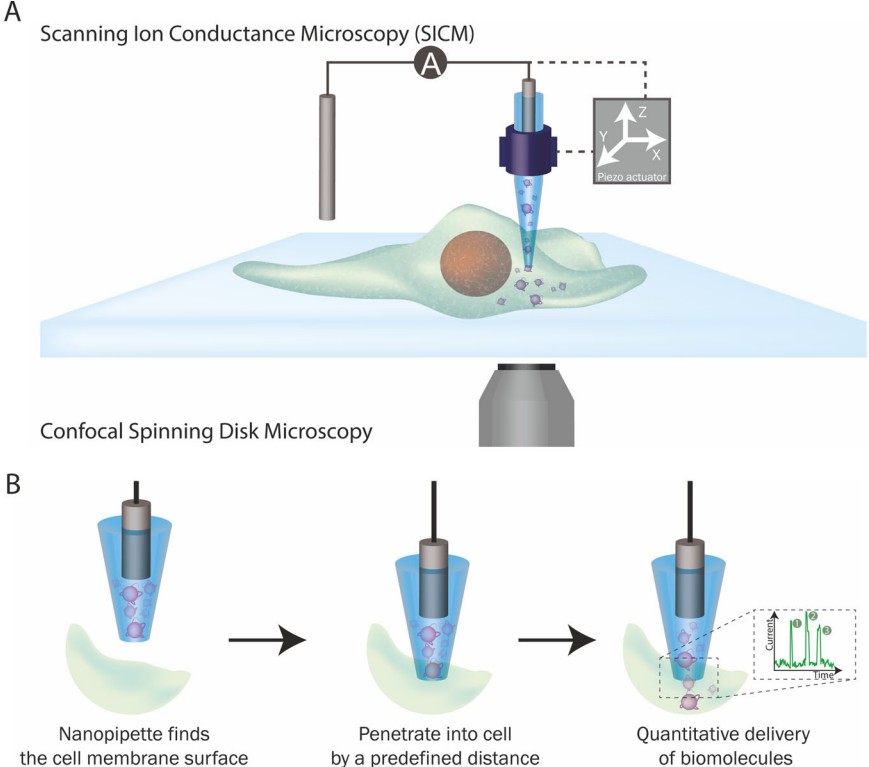

**Fig. 1 | The nanoinjection platform. A** SICM integration into the platform. The position of the nanopipette is controlled by the SICM through a piezoelectric actuator. **B** The quantitative nanoinjection procedure. The nanopipette approaches the surface of the cell membrane through the spatial control of the SICM, then the nanopipette is moved downward by a predefined distance to penetrate the cell, finally, the delivery of materials will be triggered by electrophoretic forces via the application of a suitable voltage. During delivery, the current is monitored in real-time, and the translocation of a single analyte disrupts the current baseline and appears as a peak, quantifying the number of peaks and thus revealing the number of molecules delivered to the cell.

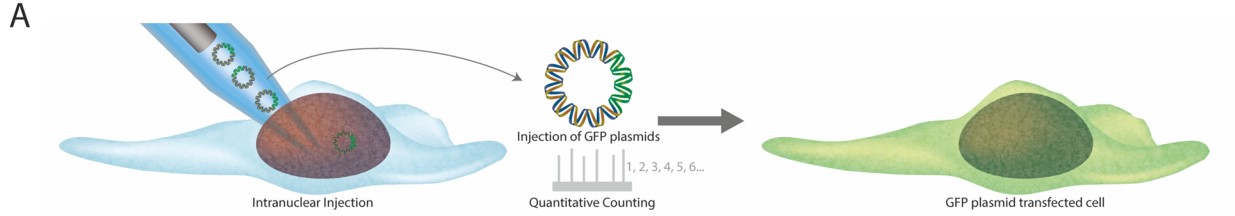

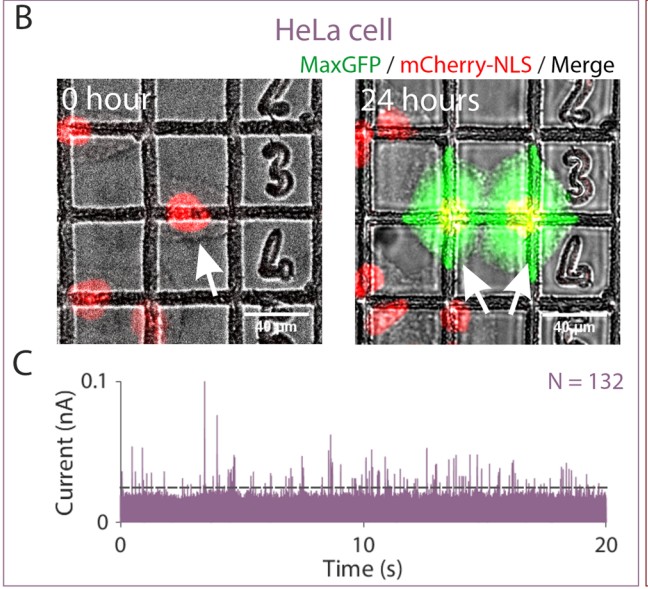

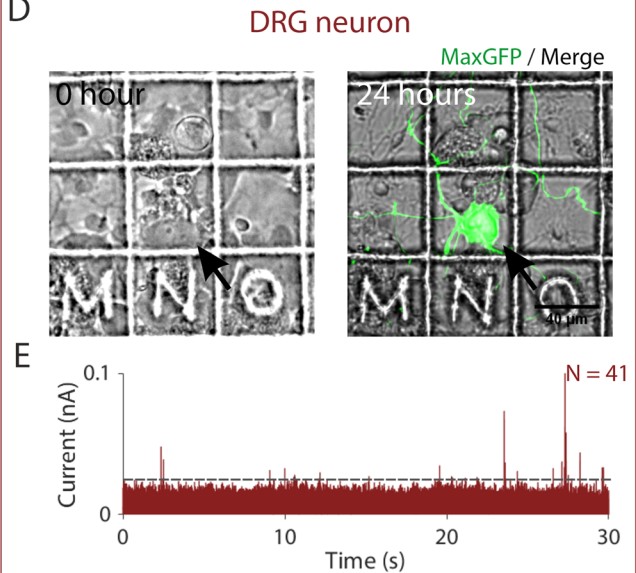

**Fig. 2 | Quantitative nanoinjection of DNA plasmids into living cells.**
**A** Schematic of the nanoinjection of GFP plasmids (pMaxGFP) into the nucleus and the transfection of the cell. **B** The transfection of a HeLa cell expressing the nuclear localised mCherry-NLS (HeLa RNuc) with pMaxGFP plasmid through quantitative nanoinjection. HeLa RNuc cells were cultured on a grided dish to enable identification of the cell after nanoinjection. pMaxGFP plasmids were quantitatively nanoinjected into the nucleus of the cell (arrow). Twenty-four hours later, the two daughter cells were imaged to confirm the expression of GFP from the injected pMaxGFP plasmids. **C** A snapshot of the current trace (20 s) recorded during the nanoinjection. Based on peak counting, a total of 132 pMaxGFP plasmids were nanoinjected into the HeLa RNuc cell. **D** The transfection of a DRG primary neuron with pMaxGFP plasmid through quantitative nanoinjection. Twenty-four hours later, the neuron was imaged to confirm the expression of GFP. **E** The current trace (20 s) was recorded during the nanoinjection step. A total of 41 pMaxGFP plasmids were delivered into the DRG neuron. The dotted line in **C** and **D** indicated the threshold for events search. The experiments were repeated three times, and the replicates can be found in the Supplementary Information. Source data are provided as a Source Data file.

can then be imaged by microscopy and/or SICM to examine the cellular effect of molecule delivery by nanoinjection after different lengths of time.

## Quantitative delivery of DNA into HeLa cells and primary neurons

Nanopipettes have been used hitherto for quantitative detection and characterisation of biological macromolecules in vitro with single molecule resolution, including globular proteins, amyloid fibrils, nucleic acids, ribosomes and nanostructures[32–39]. To demonstrate that nanopipettes can also be used for the quantitative delivery of macromolecules at single molecule resolution into cells and that this results in a demonstrable phenotypic effect, we used the nanoinjection platform to deliver a DNA plasmid into the nucleus of the human cervical epithelial HeLa cell line. We used the subsequent formation of the encoded fluorescent protein both as a readout for the successful delivery of the plasmid and for the functional integrity of the nanoinjected cell to transcribe and translate the encoded green fluorescent protein (GFP) (Fig. 2).

First, to confirm that the nanoinjection platform can perform site-specific delivery, we used the nanopipette to introduce a 70 kDa fluorescein dextran conjugate into either the nucleus or cytoplasm of HeLa cells expressing the nuclear-localised protein pmCherry-NLS (HeLa RNuc) (Supplementary Figs. 3–6). Upon insertion of the nanopipette into the cell, we observed a ~25% reduction in the baseline current (Supplementary Fig. 7). This is consistent with an increased resistance to ion flow due to the plasma membrane acting as a permeability barrier[44] and provides additional feedback to the user that the nanopipette tip is inside the cell[29,30]. The fluorescein dextran was then delivered by the application of −700 mV into either the nucleus or cytoplasm, resulting in nuclear and cytoplasmic localisation, respectively (Supplementary Figs. 5 and 6). Moreover, SICM imaging of the injection site immediately after retraction of the nanopipette revealed no residual damage to the plasma membrane, although the height of the apical plasma membrane was increased by up to ~0.5 μm (Supplementary Fig. 8).

Next, for nanoinjection of DNA, the 3.5 kbp pMaxGFP plasmid, which encodes *Pontella mimocerami* GFP, was used as a model analyte (Fig. 2A). The nanopipette containing the plasmid in a solution of phosphate-buffered saline (PBS), was inserted into the nucleus of a HeLa RNuc cell and delivery triggered by the application of −500 mV. The translocation of individual DNA molecules into the nucleus resulted in alterations of the ionic current flow through the nanopipette, with a total of 132 events being detected (Fig. 2B, C). The number of events stated corresponds to those that are detected after data processing. This may underestimate to some extent the total number of molecules that translocate through the nanopore because single molecules events with low signal to noise ratio relative to the baseline

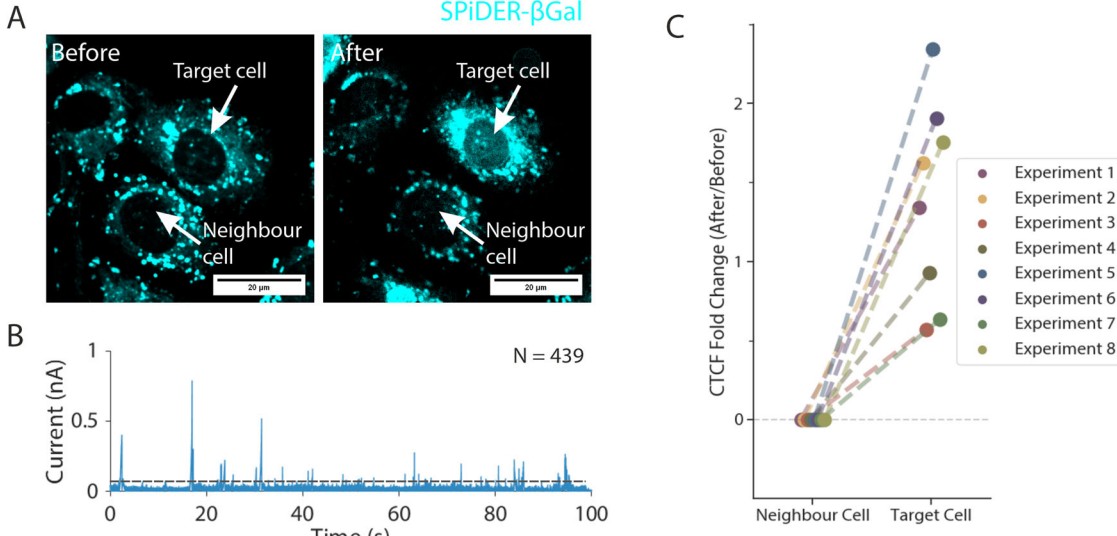

**Fig. 3 | Quantitative nanoinjection of β-galactosidase into cells. A** The dye SPiDER-βGal was used to detect β-galactosidase enzymatic activity inside the cell. Endogenous β-galactosidase is localised to lysosomes in the perinuclear cytoplasm. The nanoinjection of *E. coli* β-galactosidase into the nucleus causes the nucleus to become fluorescent. A target cell's nucleus nanoinjected with *E. coli* β-galactosidase shows an increase in nuclear overall fluorescence. **B** A snapshot of the current trace (100 s) during the nanoinjection. Based on peak counting, a total of 439 β-galactosidases were nanoinjected into the cell. **C** The Corrected Total Cell Fluorescence (CTCF) of the nucleus area before and after the nanoinjection was calculated and plotted against the molecule count for 8 independent experiments. The dotted line in **B** indicated the threshold for events search. The experiments were repeated eight times, and the replicates can be found in the Supplementary Information. Source data are provided as a Source Data file.

current would be excluded by our data analysis routine. In contrast, less than 5 events were recorded over 60 s when nanopipettes containing a mixture of PBS and ATTO 488 dye were inserted into the nuclei of HeLa RNuc cells (Supplementary Fig. 9). This is consistent with the majority of events recorded for DNA nanoinjection resulting from the translocation of the plasmid into the nucleus, although a small number of the events may be caused by the translocation of intracellular molecules into the nanopipette[45]. Twenty-four hours post nanoinjection, the HeLa cell had divided, and both daughter cells expressed GFP (Fig. 2B). This demonstrates successful delivery of the plasmid and also shows that nanoinjection is well tolerated by the cell, as least as judged by its ability to grow, divide and produce the plasmid-encoded GFP. This was replicated for two additional cells, in which 37 and 44 translocation events were detected, respectively and, for each, the cells had also divided, with both daughter cells expressing GFP (Supplementary Figs. 10 and 11). Thus, we have shown quantitative nanoinjection of DNA, with single-molecule resolution, into living cells in culture.

Whereas the transfection of immortalised cell lines, such as HeLa, is routine, the transfection of primary cells, such as neurons, can be more challenging. Using the nanoinjection platform we performed quantitative delivery of the GFP plasmid into the nuclei of two primary dorsal route ganglion (DRG) neurons (Fig. 2D, E, Supplementary Fig. 12). Forty-one and 44 translocation events were detected, respectively, and GFP expression observed in both cells 24 h later (Fig. 2D, E, Supplementary Fig. 12). Thus, the nanoinjection platform can also be used to perform quantitative delivery of plasmid DNA into primary neurons and this results in a phenotypic change with the cells expressing a protein encoded by the plasmid.

### Quantitative delivery of proteins into primary endothelial cells and neurons

The intracellular delivery of proteins, for example, CRISPR–Cas9[46,47], fluorescent proteins[48] and antibodies[49,50], into cells typically use pressure-based microinjection or electroporation[2,3]. However, proteins can misfold and aggregate under high shear pressure losing their

biological function[51], and there is little control of the number of proteins delivered into the cell when using these techniques. In previous work, we have shown that individual β-galactosidase molecules can be detected by a nanopipette when delivered into a polymer-electrolyte bath using a nanopipette[33]. Building on these observations, we used the nanoinjection platform to deliver purified *Escherichia coli* β-galactosidase into cells and to quantify the number of molecules delivered.

*E. coli* β-galactosidase is a 465 kDa tetrameric globular protein with a pI of 4.61. Hence, the protein is negatively charged at neutral pH[52]. Since *E. coli* β-galactosidase is enzymatically active only as a native tetramer[53,54], the use of this enzyme as a test substrate enabled us to determine whether the nanoinjection platform can deliver proteins without disrupting their structure and function. Mammalian cells have endogenous β-galactosidase activity in lysosomes[53,54]; therefore, to distinguish between exogenous *E. coli* β-galactosidase delivered by nanoinjection and endogenous lysosomal β-galactosidase, we performed nanoinjection of the nucleus. For these experiments, we used primary human umbilical vein endothelial cells (HUVEC), whose nuclei are easily identified because they protrude above the rest of the cell surface[55] and can thus be identified without the need for fluorescent dyes or proteins.

HUVEC cell nuclei were nanoinjected with β-galactosidase (Fig. 3A, B, Supplementary Figs. 13–16) by applying a voltage of −700 mV. The number of translocation events recorded ranged from 100 to 1000 (Fig. 3B, Supplementary Figs. 13–16). For example, for the cell in Fig. 3B, nanoinjection of β-galactosidase resulted in the detection of 439 single-molecule translocation events (Fig. 3C). β-galactosidase enzymatic activity was detected using the membrane-permeable fluorescent substrate SPiDER-βGal[56] (Fig. 3A, B, Supplementary Fig. 17 and 18), and after nanoinjection there was an increase in the fluorescence of SPiDER-βGal throughout the nucleus (Fig. 3B and Supplementary Figs. 13–16). By determining the corrected total cell fluorescence (CTCF) value for the nucleus of the injected cell and comparing it with a neighbouring non-injected cell (Methods), we confirmed that the injected cell had a greater fold increase in SPiDER-

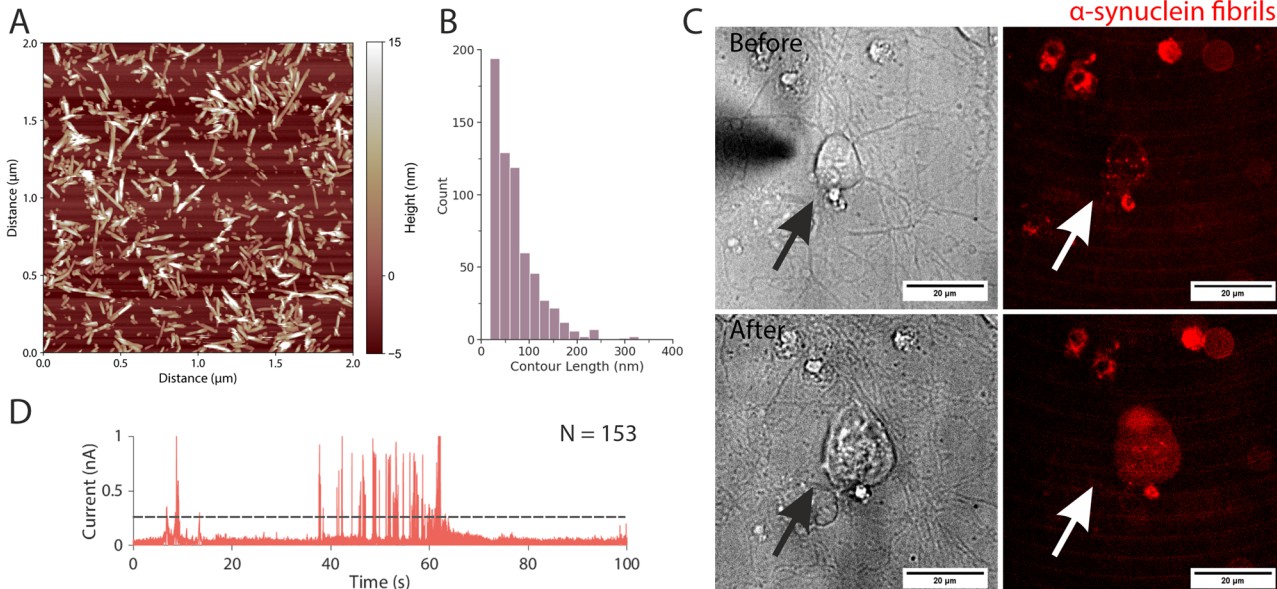

**Fig. 4 | The quantitative nanoinjection of α-synuclein fibrils into *rat* primary cortical neurons. A** A representative image of the α-synuclein fibrils and **B** their associated length distribution of 69 ± 2 nm (standard error of the mean, 628 fibrils traced). **C** The primary neuron before and after the nanoinjection of the α-synuclein fibrils. **D** A snapshot of the current trace (100 seconds) during the nanoinjection. Based on peak counting, a total of 153 α-synuclein fibrils were nanoinjected into the cell. The experiments were repeated three times, and the replicates can be found in the Supplementary Information. Source data are provided as a Source Data file.

βGal nuclear fluorescence throughout its nucleus (Fig. 3D). Conversely, no increase in SPiDER-βGal fluorescence in the nucleus was observed, relative to a neighbouring cell, when the nanoinjection platform was used to deliver a fluorescent dye into the nucleus (Supplementary Fig. 19). These data are therefore consistent with the delivery of *E. coli* β-galactosidase, in its enzymatically active form, into HUVEC nuclei.

Next, we explored whether nanoinjection platform can perform quantitative delivery of protein fibrils into cells. For this, we used α-synuclein amyloid fibrils generated in vitro (Methods)[57]. We have shown previously, by using a polymer-electrolyte bath, that delivery of α-synuclein amyloid fibrils can be detected by a nanopipette[33]. To enable visualisation of the fibrils delivered into cells, the A90C α-synuclein fibrils were labelled with Alexa Fluor 594 on the cysteine residue (Methods). AFM imaging revealed that the labelled fibrils had an average length of 69 ± 2 nm (Fig. 4A, B, Supplementary Fig. 20). The fibrils were nanoinjected into the cytoplasm of primary rat cortical neurons (Fig. 4C and Supplementary Figs. 21 and 22). In the example shown in Fig. 4C, 628 events were detected (Fig. 4D), and this resulted in a corresponding increase in Alexa Fluor 594 fluorescence in the injected neuron (Fig. 4C). Two additional neurons were nanoinjected with 305 and 426 events detected, respectively, and the neurons were observed to have increased cell-associated fluorescence after the injection (Supplementary Figs. 21 and 22). Thus, we have demonstrated that nanoinjection can perform quantitative delivery, with single-molecule resolution, of fibrous proteins into primary cells.

**Effects of the intracellular environment on single molecule translocation**

We have shown previously that the signal-to-noise ratio (SNR) for the detection of molecules by nanopipettes is dependent on the composition of the electrolyte solution into which the molecule is translocated and can be enhanced by using the synthetic polymer PEG[33,36–38]. The intracellular environment is a complex mixture of macromolecules, small molecules and ions, densely packed and crowded, and hence is very different to the electrolyte baths typically used for single molecule detection by nanopipettes[58]. We, therefore, investigated whether the intracellular environment affects the ionic current signatures of the single molecule translocations. For this, we compared the translocation of a model analyte, a linear 7kbp dsDNA molecule (Methods), delivered into cells with that of the DNA translocated into an electrolyte bath of either the electrolyte solution PBS or PBS containing 30% w/v bovine serum albumin (PBS BSA), as a simple mimic of the intracellular crowded environment[59] (Fig. 5).

The DNA was delivered sequentially into a HeLa RNuc cell, into a PBS electrolyte bath, and then finally into a 30% (w/v) BSA PBS bath (Fig. 5A). Crucially, by using the same nanopipette for all three conditions, we could discount any differences in the nanopipette's geometry and dimensions on the translocation signatures of the DNA. To confirm the cell had been injected, the nanopipette also contained the small molecule fluorescent dye ATTO 488. Translocation events were detected for the DNA in each instance (Fig. 5B and Supplementary Fig. 23–26). An increase in both the current peak amplitude and the dwell time for the events recorded for the delivery of the DNA into either the cell or 30% (w/v) BSA PBS bath was observed compared with delivery into PBS (Fig. 5C). This was reflected in the integrals of the area calculated for each translocation event, which showed a shift from under 100 fC for PBS, to close to ~200 fC for the cell and PBS-BSA (Fig. 5D). These results imply that the cellular environment enhances the detection of DNA into cells, and that macromolecular crowding in the cell maybe responsible for the increased SNR.

To gain further insight into the effects of macromolecular crowding on the translocation of DNA, we employed a coarse-grained molecular dynamics method to simulate the moment a DNA molecule reached the nanopore at the tip of the nanopipette, and its translocation into a standard electrolyte environment versus a macromolecular crowded environment composed of 30% (w/v) BSA. The simulations described the translocation of 2.7 kbp DNA molecules through a model of a nanopipette having a 10 nm aperture, with and without Lennard−Jones particles representing BSA molecules under a −600 mV applied bias (Fig. 6A). The ionic current was estimated from the translocation trajectories, allowing an enhancement relative to the open-pore current to be computed (Fig. 6B). Each simulation ensemble, with and without BSA molecules, consisted of 24 independent runs (Supplementary Movies 1−6). For each simulation,

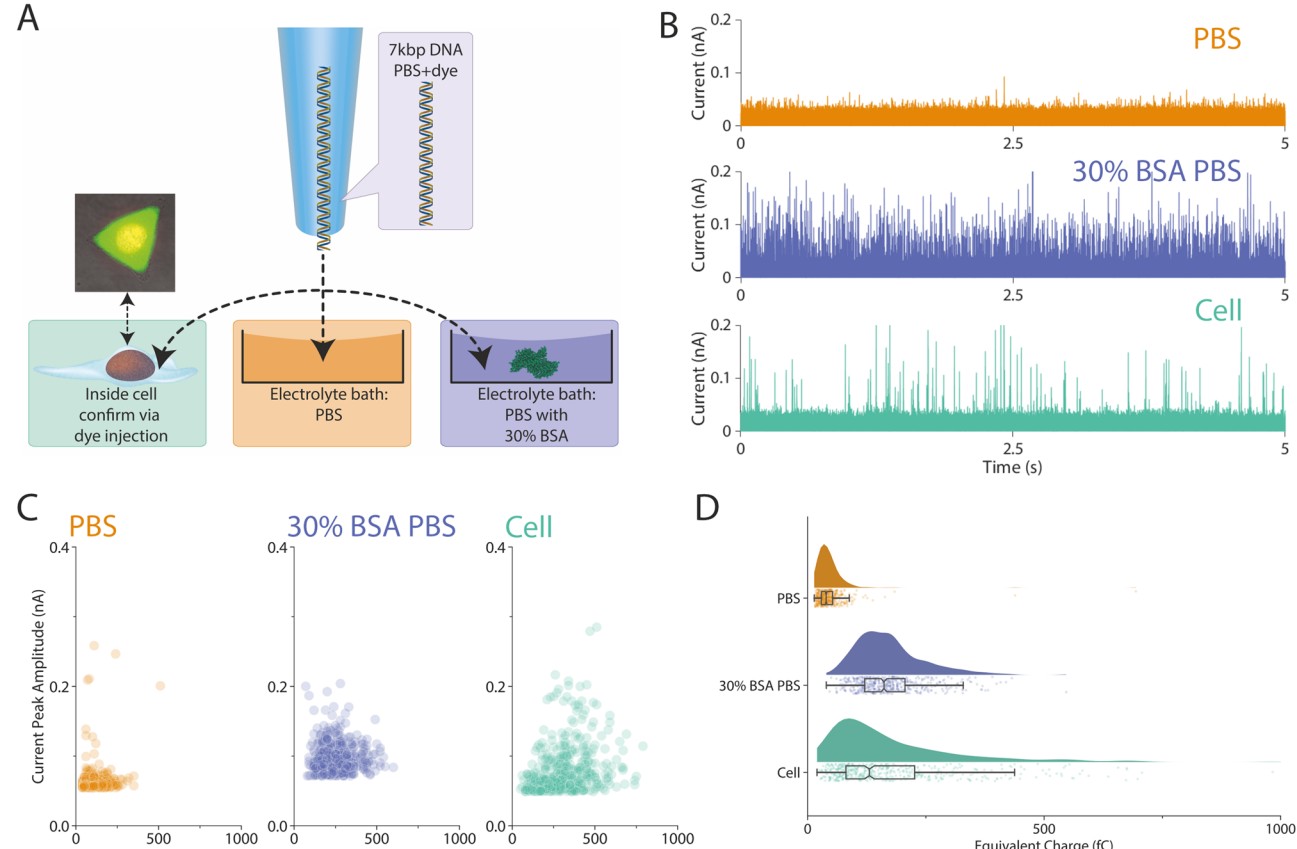

**Fig. 5 | Analysis of the effects of the intracellular environment on single molecule translocation of DNA. A** The same nanopipette was filled with 5 nM 7 kbp dsDNA in PBS mixed with 10 μM ATTO 488 fluorescent dye, and a voltage of −500 mV was used to drive the dsDNA from the nanopipette to either the HeLa RNuc cell, PBS or 30% (w/v) BSA PBS. The cell turns fluorescently green after the injection due to the ATTO 488. **B** The 5 s current traces of the translocation of the dsDNA into either PBS, 30% (w/v) BSA PBS or cell. **C** The population distribution of the translocation event of the 7 kbp dsDNA, both 30% (w/v) BSA PBS and cell show a wider distribution on the dwell time. **D** The equivalent charge of the translocation events was plotted, and a clear shift can be observed between PBS and 30% (w/v) BSA PBS and cell. The box and whisker plots show the median value, the 25th and 75th percentile (box) and upper extreme and lower extreme (whisker). For **C** and **D**, a total of 500 events were randomly sampled and plotted. The experiments were repeated three times, and the replicates can be found in the Supplementary Information. Source data are provided as a Source Data file.

the translocation process could be monitored (Fig. 6C), and the elapsed time between the first and last DNA beads exiting the pore was obtained (Fig. 6D), revealing that the presence of BSA molecules slowed the translocation of the DNA and increased the current.

The simulations allowed qualitative investigation of multiple factors that cause the current enhancement during DNA translocation in the presence of BSA at the aperture of the nanopore opening. We observed that the DNA was initially more compact as it was being translocated in either the presence or absence of BSA, as characterised by its radius of gyration. By the end of the translocation process, the DNA swelled ~10% more in the absence of BSA than it did when translocated in the presence of BSA (Fig. 6E). Moreover, in the presence of BSA, the DNA continued to be spread out more slowly after the last base pair exited the pore, likely contributing to a slower recovery of the current towards the baseline (Fig. 6F).

The enhancement in the ionic current caused by BSA can be explained by two mechanisms. First, due to the closer proximity of the DNA to the nanopore aperture, the DNA directly increases the ionic current through a direct effect on the conductivity of the solution near the nanopore[60,61]. Second, the DNA displaces a small number of BSA molecules near the nanopore aperture that would otherwise be sterically blocking the ionic current flow, thus resulting in an increase in the ionic current.

## Discussion

Herein we demonstrate the development of a nanoinjection platform in which a nanopipette is used both as an SICM scanning probe and as an injection probe. We describe the application of this device in the precise manipulation of living cells, demonstrating that the platform can perform the delivery of DNA, globular proteins, and protein fibrils into different cellular locations with single molecule resolution. Moreover, we demonstrate that the injection process is well tolerated by different cell lines and primary neurons and show that the delivery of biological macromolecules into cells can result in a demonstrable phenotypic change.

Several different approaches have been developed over recent years using nanoscale probes for the delivery of materials into cells[3]. Examples include using hollow nanoelectrodes for the intracellular delivery of individual gold nanoparticles that could be monitored by enhanced Raman scattering[62]. Solid-state nanopores coupled with optical tweezers have also been used for proof-of-concept experiments in single-cell transfection, but did not demonstrate protein injection or the ability to manipulate primary cells[63]. Pandey et al. employed a multifunctional nanopipette for the intracellular delivery of single entities, including a model protein, ferritin, and PEGylated gold nanoparticles[64]. Our study provides a direct demonstration that biological macromolecules can be delivered intact into defined cellular locations (herein nucleus vs cytoplasm) visualised by expression of

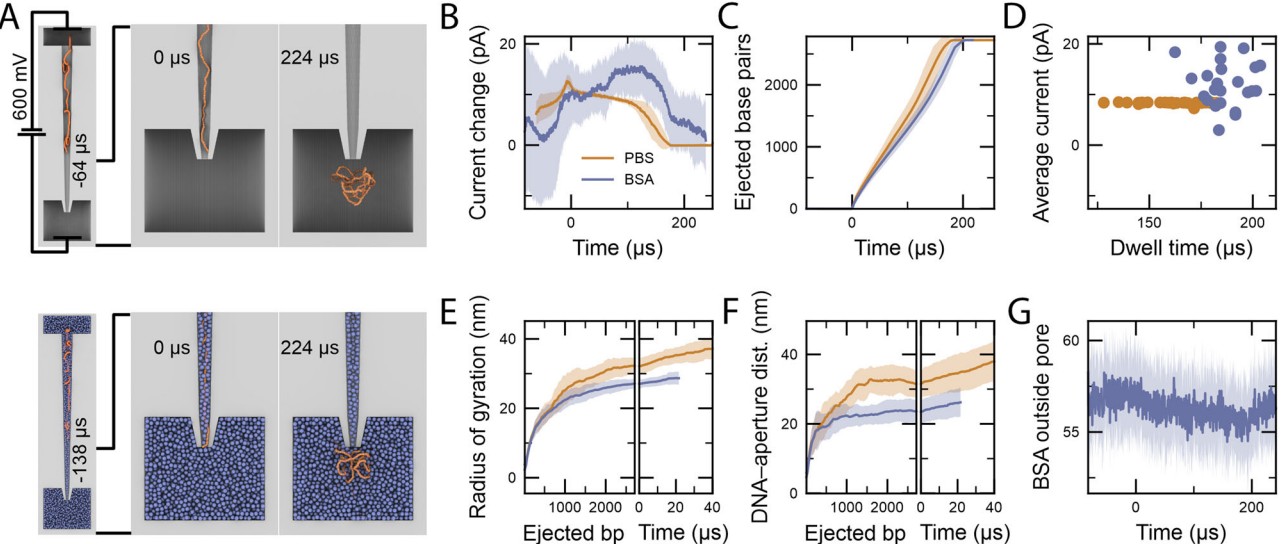

**Fig. 6 | Coarse-grained simulations of DNA translocation. A** Coarse-grained simulation systems consisting of a 2.7 kbp dsDNA molecule (orange) driven out of a nanopipette (grey) by an applied electric potential into an electrolyte solution with and without BSA proteins (blue). Each simulation ensemble (with or without BSA) consisted of 24 independent runs. **B** Ionic current enhancement as the DNA moved through the pore, averaged over each simulation ensemble. Here and throughout the figure, solid lines depict the ensemble average, whereas shaded regions depict the standard deviation among the simulations. **C** Number of base pairs having left the pore during the simulations averaged over each ensemble. **D** Scatter plot showing the elapsed time between the first and last base pair being translocated through the pore in each simulation against the average current enhancement during that time interval. **E** Radius of gyration of DNA having been translocated through the pore plotted against the number of translocated base pairs (left) or the time since the last base pair was translocated (right). **F** Distance of the centre of mass of the ejected DNA from the pore aperture, projected along the pore axis and plotted against the number of translocated base pairs (left) or the time since the last base pair was translocated (right). **G** Number of BSA molecules below the aperture. The number of molecules was analysed in a cylindrical volume sharing the axis of the pore and immediately below the aperture with a 15 nm radius and 30 nm height. The shadings represent the s.d. calculated for 24 independent simulations. Source data are provided as a Source Data file.

plasmid-encoded genes, detection of β-galactosidase enzymatic activity and imaging of fluorescent α-synuclein fibrils. Crucially, the finding that cells divide after injection, produce plasmid-encoded GFP and enable detection of enzymatically active β-galactosidase demonstrate that cellular function is not perturbed and that protein structure is not affected by the nanoinjection procedure.

We demonstrate intracellular delivery of plasmid DNA, β-galactosidase and α-synuclein amyloid fibrils, all of which we have shown previously to be detected by nanopipette when translocated into a polymer-electrolyte bath[33]. It also potentially opens the door for detailed investigation into concepts such as the proteotoxicity associated with amyloid fibril formation and screening for the effects of small molecules or molecular chaperones on the cellular proteotoxicity of amyloid aggregates. Given that the self-assembly of α-synuclein and other proteins involved in neurodegeneration occur intracellularly[65], the introduction of preformed fibrils into neurons offers a route to determine directly the effects of protein fibrils on cellular homoeostasis to compare with results obtained by the commonly adopted practice of adding preformed fibrils to culture medium[66–70]. Nanoinjection of α-synuclein aggregates into neurons also enables quantification of the number of aggregates delivered, enabling the direct comparison of the effects of different numbers of fibrils on cellular response. Finally, and importantly, given that different fibril structures are associated specifically with different diseases, even when formed from very similar or even identical sequences[71], the nanoinjection platform developed here will enable the cellular effects of individual fibril types to be directly and quantitatively compared.

In its current form, the nanoinjection platform has some scalability limitations that restrict its throughput to about 10 cells per hour (if all cells are injected with the same nanopipette). This is due to the time required (i) to identify target cells by light microscopy, (ii) for the SICM to approach and position the nanopipette over the cell, and (iii) for the penetration of the nanopipette into the cells

and delivery of macromolecules. The speed of some of these steps could, however, be increased, for example, target cell identification, cell penetration and injection can be refined through custom software and hardware combinations, as used in recent advances in automating microinjection [61–63] The response time of the SICM hopping mode could also be improved, as shown recently with the development of adaptive hopping mode[64]. However, despite its low throughput, nanoinjection has some significant advantages. It can deliver molecules directly into the cell and, as appropriate, into the nucleus, and by monitoring translocation events, nanoinjection can provide direct feedback to the user that molecules have been delivered successfully. As such, using DNA as an example, the efficiency of transfection for each nanoinjected cell will be high, and we also show that it can be used for 'hard' to transfect cells, such as primary neurons[72].

In addition to macromolecule delivery nanoinjection will result in the transfer of solution from the nanopipette into the cell. The nanopipettes used for the nanoinjection herein have a diameter of c.a. 24 nm (Supplementary Fig. 1B). The volume delivered can be estimated, according to Babakinejad et al., by calculating the flow rate[65]. We estimate that the total flow rate of a c.a. 24 nm wide nanopipette would be 33 fl/s. Thus, approximately 2 pl of solution will be delivered per minute. Based on this, the transfection of HeLa cells and primary neurons through nanoinjection would have resulted in ~0.5 pl of solution being delivered into the cell (assuming a fluid flow rate of 33 fl/s). It is important to note that the estimation of flow rate is based on delivery into 1× PBS solution[65], and the flow rate inside the cell may be different. Moreover, no evidence of an adverse effect on cell viability was observed, as the injected cells all expressed GFP and the HeLa cells divided. Consistent with this observation, studies using the FluidFM technology have shown that introducing near 1 pl into a HeLa cell does not affect the cell viability[8,66,67], and similarly, the removal of 4 pl of cell cytosol has negligible effects on cell viability[68].

One intriguing observation we report here is that the translocation of macromolecules into the cell increases the sensitivity of the nanopipette, with an enhanced SNR compared to the delivery of macromolecules into an electrolyte bath. This signal enhancement was also observed for an electrolyte bath containing BSA, suggesting that the SNR is increased due to macromolecular crowding in the cell. For translocation into a macromolecular crowded BSA bath, the coarse-grained simulation showed that the DNA, after translocation, displaced BSA molecules around the nanopore and that the DNA remained near the nanopipette opening, resulting in an increase in the ionic current flow. We propose that a similar phenomenon is likely for the observed SNR enhancement for the translocation of DNA and protein molecules from the nanopipette into cells, with macromolecular crowding being responsible, at least in part, for the increased SNR, thus enhancing the sensitivity of the instrument for use in cells.

In summary, we demonstrate a nanoinjection platform for the quantitative delivery of macromolecules into both cell lines and primary cells with single-molecule resolution. The platform is universal and could also enable the injection of ribosomes, DNA origami and viral RNAs[34,36,37] for precise cellular manipulation. We also believe this approach will provide a wide range of applications for the nanoinjection platform, enabling new insights into structure-function relationships of protein and protein complexes in the cell.

## Methods

### Fabrication of nanopipettes

The nanopipettes were fabricated from quartz capillaries of 1.0 mm outer diameter and 0.5 mm inner diameter (QF100-50-7.5; Sutter Instrument) using the SU-P2000 laser puller (World Precision Instruments). A two-line protocol was used: (1) HEAT 750/FIL 4/VEL 30/DEL 150/PUL 80, (2) HEAT 700/FIL 3/VEL 40/DEL 135/PUL 149. The polling protocol is instrument-specific and can vary between different pullers.

Details about the nanoinjection, cell culture protocols, and characterisation of the nanopipettes, are included in the Supplementary Information.

### Scanning ion conductance microscopy and spinning disk confocal microscopy

The SICM scan head consisted of a Z-piezo motor with a range of 25 μm for the vertical positioning of the nanopipette and a 100 μm XY-piezo motor for the lateral positioning of the sample (Ionscope). The SICM set-up utilises the AxoPatch 200B (Molecular Devices) patch-clamp amplifier in voltage–clamp mode. The signal was filtered using a Bessel filter at 10 kHz and digitised with a Digidata 1440 A (Molecular Devices) at a 100 kHz (interval 10 μs) sampling rate and recorded using the software pClamp 10 (Molecular Devices). The HPICM software (Version 1.3.0.11, ICAPPIC Ltd.) was used to control the SICM set-up[42]. A nanopipette filled with analyte was used to approach the cell surface via the hopping mode, where the nanopipette is vertically approached to the cell surface until the ion current drops below 99.5% of the baseline ion current, thereby defining the height of the surface at this position[42]. Repeating this procedure for many positions on the cell surface generates an image of cell topography. The SICM image data was processed using the HPICM software (Version 1.3.0.11, ICAPPIC Ltd.) was used to. All SICM experiments required the use of a 35 mm glass bottom culture dish.

The SICM experiments on cells were performed in $CO_2$-independent Leibovitz's L-15 medium (21083-027; Gibco). The SICM set-up is placed directly on top of a Yokogawa Spinning Disk confocal system coupled with the ANDOR iQ3 live cell imaging system (Oxford Instruments), allowing fluorescence and brightfield imaging. The confocal microscope was fitted with a 455, 488 and 561 nm laser and emission filter set that enables the visualisation of a wide range of fluorescent dyes. The nanopipette tip was aligned with the microscope and

positioned next to a cell of interest for scanning or nanoinjection. Unless stated otherwise, all fluorescent images were captured by the ANDOR iQ3 live cell imaging system with appropriate excitation laser and emission filter combinations.

### Nanoinjection

For all nanoinjection procedures, the ion current was recorded by pClamp 10 (Molecular Devices). All fluorescent images were captured by the ANDOR iQ3 live cell imaging system with appropriate excitation laser and emission filter combinations. The nanopipette was lowered to 10 μm/s during cell penetration. Detailed information on the composition of the nanoinjection analyte can be found in the Supplementary information.

### Single-molecule detection

For the translocation experiments, the nanopipettes were filled with analyte of interest diluted into 1× PBS, the nanopipette was fitted with an Ag/AgCl working electrode. The tip of the nanopipette was then immersed into the electrolyte bath of choice with a grounded Ag/AgCl reference electrode, thus establishing a complete electric circuit between the inside of the nanopipette to the outer bath solution. Depending on the polarity of the analyte, the application of a voltage to the working electrode caused molecules from inside of the nanopipette to translocate through the nanopore and into the bath solution. The ionic current was measured using a Multi-Clamp 700B (Molecular Devices) patch–clamp amplifier in voltage–clamp mode. Unless specified, the signal was filtered using a Bessel filter at 10 kHz and digitised with a Digidata 1550B (Molecular Devices) at a 100 kHz sampling rate (every 10 μs) and recorded using the software pClamp 10 (Molecular Devices). Translocation event current analysis was carried out with a custom MATLAB script (provided by Prof Joshua B. Edel, Imperial College, London, UK). The MATLAB script identifies individual events in a given ion current trace using defined thresholds, at least five standard deviations above baseline noise. The baseline is tracked via an asymmetric least square smoothing algorithm, and the fit is determined by the Poisson probability distribution function.

### Coarse-grained simulation

The coarse-grained mrDNA model[73] was used to simulate the translocation of linear 2.7 kbp dsDNA molecules through a nanopipette represented by a grid-based potential using the ARBD simulation engine (version Feb22)[74]. Twenty-four simulations were performed per solvent condition–with or without Lennard-Jones spheres (3.9 nm $R_{min}$; 0.1 kcal mol$^{-1}$ $\epsilon$) at a concentration of 4.5 mM representing crowding BSA molecules. DNA–BSA interactions were computed by attributing by setting $R_{min}$ to 1.1 nm for DNA beads and $\epsilon$ to $0.05 \times N_{nt}$ kcal mol$^{-1}$ where $N_{nt}$ is the number of nucleotides represented by a DNA bead. BSA molecules were assigned a damping coefficient of 215 ns$^{-1}$, and a 40 fs timestep was used to advance the configuration of the system while Langevin forces maintained a 291 K temperature. In all simulations, the nanopore was represented by the electrostatic potential obtained from a previously described[75] finite element COMSOL model of a nanopipette, with geometry adapted to a 10-nm-diameter aperture and a 600 mV applied electrostatic potential ejecting the DNA from the pipette. The electrostatic COMSOL axisymmetric solution was sampled at regular lattice sites in cylindrical coordinates before being interpolated onto a 3D grid using a custom Python script. A steric grid potential to prevent the DNA from entering the pore walls was obtained using a custom Python script to compute the distance $d$ of a given voxel from the nearest voxel with a valid solution, setting the steric potential at a given voxel to $\frac{1}{2}kd^2$, where $k = 2$ kcal mol$^{-1}$ Å$^{-2}$. The steric potential acting on the centre of BSA spheres was obtained by assigning voxels located out of the COMSOL domain a value of 10 kcal per mol (zero elsewhere) and convolving with

a normalised 3D kernel representing the size of the BSA that consisted of a linear ramp ranging from zero to one for distances 2.5 to 4.5 nm from its center. The harmonic steric potential applied to the DNA was added to the steric potential acting on BSA. The initial DNA configurations were obtained from previously performed simulations with the voltage reversed. The twenty-four simulations in each ensemble consisted of three subgroups of eight simulations, each with the DNA end nearest the aperture initially placed around 100, 185, or 270 nm. BSA beads were initially randomly distributed through the system.

Each simulation system was equilibrated for 50–100 µs in the presence of the steric potentials and in the absence of an electrostatic potential. After equilibration, the electrostatic potential was turned on until the DNA was fully translocated from the nanopipette. A steric exclusion model (SEM)[76] was used to process each trajectory to obtain an estimate of the ionic current as previously described[75] using data obtained from atomistic simulations of the monovalent ion enhancement around DNA to estimate the associated current enhancement near a DNA molecule in 170 mM KCl solution. Before calculating the ionic current, the ionic mobility map, including the DNA enhancement, was modulated by a BSA-distance-dependent function. Briefly, at each site in a discretized grid, the distance to the nearest BSA molecule was computed, and the mobility was scaled by a linear ramp from zero to one between distances of 2.5 and 4 nm. Eight simulations lasting 80–100 µs each were used to estimate the bulk ion conductance of the BSA solution in the absence of DNA and used to compute the modulated current.

## Statistics and reproducibility
A number of technical replicates are defined in the legends of the figures, and the data of the technical replicates can be found in the Supplementary information. The box and whisker plots of Fig. 5D show the median value, the 25th and 75th percentile (box) and upper extreme and lower extreme (whisker). The shadings in Fig. 5 represent the s.d. calculated for 24 independent simulations. Statistical analyses were performed with Python (version 3.9). No statistical method was used to predetermine the sample size.

Supplementary movies showing the simulation of the DNA translocating from the nanopipette to the electrolyte bath without BSA (Supplementary Movies 1, 3, and 5) or with BSA (Supplementary Movies 2, 4, and 6):

## Reporting summary
Further information on research design is available in the Nature Portfolio Reporting Summary linked to this article.

## Data availability
Data (ionic current traces, fluorescent micrographs) supporting this work can be freely accessed via the University of Leeds data repository: https://doi.org/10.5518/1512. Additional relevant information is available from the corresponding author. Source data are provided with this paper in the Source Data file.

## Code availability
All ionic current traces were analysed using a custom-written Matlab script provided by Prof Joshua Edel (Imperial College London). Request to access the script should be addressed directly to Prof Edel at https://www.imperial.ac.uk/people/joshua.edel.

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

## Acknowledgements

We thank Prof Nikita Gamper of the University of Leeds for providing the DRG neurons. We thank Prof Derek Steele of the University of Leeds for providing technical support and access to the spinning disk confocal microscope. We thank Prof Joshua Edel for sharing the nanopore data analysis script with us. We thank Dr Michael Davies for providing the fluorescently labelled fibrils. C.C. and P.A. acknowledge funding from the Engineering and Physical Sciences Research Council UK (EPSRC) Healthcare Technologies for the grant EP/W004933/1. C.C. and P.A. acknowledge funding from the Biotechnology and Biological Sciences Research Council (BBSRC) for the Research Grant BB/X003086/1. E.H. and S.E.R. acknowledge funding from the Wellcome Trust for the Research Grant 094266/Z/10/Z. S.E.R. holds a Royal Society Professorial Fellowship (RSRP/R1/211057). This work was supported by the Human Frontier Science Project (RGP0047/2020) and the Leverhulme Visiting Professorship grant [VP2-2019-012] to A.A. ARBD development is supported by the National Science Foundation grant OAC-2311550. The supercomputer time was provided through the Leadership Resource allocation MCB20012 on Frontera of the Texas Advanced Computing Center and the ACCESS allocation MCA05S028. For the purpose of Open Access, the authors have applied a CC BY public copyright licence to any author accepted manuscript version arising from this submission.

## Author contributions

C.C. co-designed the research study, performed the experiments, and analysed data. C.C. illustrated all schematics. A.A., P.A., S.E.R. and E.H. co-designed the study, supported the data analysis, supervised the work and acquired the funding. C.M. performed the simulations and interpreted the data. All authors contributed to the writing of the paper.

## Competing interests

The authors declare no competing interests

## Ethics

All animal work carried out at the University of Leeds was approved by the University of Leeds Animal Welfare Ethical Review Body (AWERB) and performed under UK Home Office License P40AD29D7 to Nikita Gamper and in accordance with the regulations of the UK Animals (Scientific Procedures) Act 1986.
