## [Peer Review File · Nature Communications]

Single molecule delivery into living cellsREVIEWER COMMENTS

Reviewer #1 (Remarks to the Author):

General Comments to the paper:

The authors are pioneering new route in ex vivo cellular engineering, addressing a critical challenge: the manipulation of cells without inducing irreversible harm or compromising their functionality. The manuscript demonstrates that their innovative nanoinjection platform, which incorporates a nanopipette within the SICM, facilitates safe, quantised, effective, and non-destructive penetration into the intracellular space across a variety of cell types. This technology holds promise for advancing cellular engineering by enabling precise intracellular interventions without damaging the cells.

Overall the experimental and the SI are very convincing, and robust set of data is being presented. I really like the scheme and the various steps that are describing the approach (Figure 1 a, b).

Specific comments:

1. In the introduction the authors failed to mention the use of nano needles as an effective route for nano injection as well as nanobiospy. Then the he following references show be added in the last sentence of the first paragraph

[1] Nature Nanotechnology 17, 807–811 (2022);

[2] Nature Rev Mater 7, 953–973 (2022);

[3] Nature Protocol 2021, 16, 4539;

2. In the second paragraph (last sentence) the authors should insert in their list of approaches: that polymeric and silicon nano needles and their electro active analogues as well as appropriate references including : (i)Adv Funct Mater, 2022, 32, 2104828;

(ii) Adv Mater 2023, 35, 2304122; Nanobiotechnol 21, 273, 2023

3. page 3 (Word file), third paragraph "suitable voltage"—could the authors provide with range of values, along with elaboration of how those values are being chosen for the direct translocation of diverse range of cargoes—regardless of their type, size, and surface charge?

4. In the introduction, the authors should address the scalability limitations of this approach.

5. "the translocation of individual DNA molecules into the nucleus resulted in alterations of the ionic current flow through the nanopipette, with a total of 132 events" is that mean that 132 transactions events occurs of plasmid of insertion? The authors should clarify this point right at the get go.

6. How do the authors overcome the migration challenges over the tested time of the cells they are probing?

Reviewer #2 (Remarks to the Author):

Authors claimed that they developed a single molecule delivery into a living cell by using scanning ion conductance microscopy.

Based on the shown result, it is believed that quantitative delivery into a live cell is performed.

However, if it is considered what the final goal of the delivery is, it is very difficult for me to agree that showing the quantitative delivery itself is enough.

First of all, I think they need to show how fast this delivery can be done, and how many cells can be used for this delivery during a certain amount of time.

In addition, how much volume can be delivered into a single cell., and how the cell viability will be changed with respect to injected volume or molecules.

Finally, I wonder how efficiently this method can perform tranfection compared with conventional methods.

I think this manuscript will be very strong with these additional result that I suggest.

REVIEWER COMMENTS

Reviewer #1 (Remarks to the Author):

General Comments to the paper:

The authors are pioneering new route in ex vivo cellular engineering, addressing a critical challenge: the manipulation of cells without inducing irreversible harm or compromising their functionality. The manuscript demonstrates that their innovative nanoinjection platform, which incorporates a nanopipette within the SICM, facilitates safe, quantised, effective, and non-destructive penetration into the intracellular space across a variety of cell types. This technology holds promise for advancing cellular engineering by enabling precise intracellular interventions without damaging the cells. Overall the experimental and the SI are very convincing, and robust set of data is being presented. I really like the scheme and the various steps that are describing the approach (Figure 1 a, b).

Specific comments:

1. In the introduction the authors failed to mention the use of nano needles as an effective route for nano injection as well as nanobiopsy. Then the he following references show be added in the last sentence of the first paragraph

[1] Nature Nanotechnology 17, 807–811 (2022);

[2] Nature Rev Mater 7, 953–973 (2022);

[3] Nature Protocol 2021, 16, 4539;

We have now included reference to the nanoneedles works by including the suggested references to the paper.

2. In the second paragraph (last sentence) the authors should insert in their list of approaches: that polymeric and silicon nano needles and their electro active analogues as well as appropriate references including : (i)Adv Funct Mater, 2022, 32, 2104828; (ii) Adv Mater 2023, 35, 2304122; Nanobiotechnol 21, 273, 2023

We thank the reviewer for pointing this out and we have now added these references in the introduction of the manuscript.

3. page 3 (Word file), third paragraph "suitable voltage"—could the authors provide with range of values, along with elaboration of how those values are being chosen for the direct translocation of diverse range of cargoes—regardless of their type, size, and surface charge?

We have added an additional paragraph in the introduction to clarify that “suitable voltage” will depend on molecule’s charge because delivery utilizes electrophoretic forces.

4. In the introduction, the authors should address the scalability limitations of this approach.

We have added an additional paragraph to the Discussion to address the scalability of this approach and suggested approaches to improve the throughput of the nanoinjection platform.

5. "the translocation of individual DNA molecules into the nucleus resulted in alterations of the ionic current flow through the nanopipette, with a total of 132 events" is that mean that 132 transactions events occurs of plasmid of insertion? The authors should clarify this point right at the get go.

The translocation events detected were after data processing, as such we cannot rule out that there are translocations that are not detected, for example if a molecule moves too fast and exceeds our instrument limits or if a translocation event has a low current signal in relative to background noise, and thus cannot be distinguished from noise. We state these caveats in the Results section when describing the data from Figure 2 and define the events counted as corresponding to those after data processing. In addition, it is important to note, we show that the cell itself increases the SNR for translocation events and this will increase the proportion of translocation events that we can detect.

6. How do the authors overcome the migration challenges over the tested time of the cells they are probing?

For DNA nanoinjection, the cells were plated on a grided dishes (<https://ibidi.com/gridded-dishes/177--dish-35-mm-high-grid-50-glass-bottom.html>) as can be seen from Figure 2. The grided dishes allowed us to monitor the same cell the next day. While cells can migrate to some extent, we were able to use the grid reference to identify the cell the next day. We observed that the nanoinjected cells were located to the same grid coordinates the next day as confirmed by expression of GFP. Alternative approach that could be applied are the delivery of fluorescent dye labelled molecules (e.g. fibrils in Figure 4) or dye co-injection (S.Figures 23 - 26).

Reviewer #2 (Remarks to the Author):

Authors claimed that they developed a single molecule delivery into a living cell by using scanning ion conductance microscopy.

Based on the shown result, it is believed that quantitative delivery into a live cell is performed.

However, if it is considered what the final goal of the delivery is, it is very difficult for me to agree that showing the quantitative delivery itself is enough.

The overarching objective of this manuscript is to describe and validate a universal nanoinjection platform able to deliver a range of biological molecules (DNA, proteins and protein aggregates) into living cells with single molecules precision. We fully agree with the reviewer that future work needs to be focused on answering biological questions and showing the advantages of quantitative delivery and we presented some of these questions in the Discussion section. In the revised manuscript. We have also included a further reference to a recent commentary Nature Reviews Molecular Cell biology (<https://www.nature.com/articles/s41580-023-00627-6>) which highlights how transfection is still a poorly understood and insufficiently characterized technique and the need for quantitative tools for the characterization of the transfection process.

First of all, I think they need to show how fast this delivery can be done, and how many cells can be used for this delivery during a certain amount of time.

This comment has been addressed by including a paragraph in the discussion, as a similar comment was raised by reviewer 1.

In addition, how much volume can be delivered into a single cell., and how the cell viability will be changed with respect to injected volume or molecules.

We have now included a paragraph in the Discussion to address this point. To summarise, for DNA nanoinjection in figure 2, we estimate that ~0.5pL would be delivered according to the flow rate calculated by Babakinejad et al. The expression of GFP the next day indicates no adverse effect on cell viability, and consistent with this it has been shown for Fluid FM introduction of near 1 pL does not affect cell viability.

Finally, I wonder how efficiently this method can perform tranfection compared with conventional methods.

As already addressed for reviewer 1 in a paragraph added to the Discussion, compared to conventional transfection methods, nanoinjection is limited by scalability and has a lower throughput. However, it can deliver molecules directly into the cell, and as appropriate into the nucleus, and by monitoring translocation events nanoinjection can provide direct feedback to the user that molecules have been delivered successfully into the cell. As such the efficiency of transfection for each nanoinjected cell will be high compared to other methods and we also show that it can be used for 'hard' to transfect cells, such as primary neurons. Additional text added to the Discussion to address this point

I think this manuscript will be very strong with these additional result that I suggest.

REVIEWERS' COMMENTS

Reviewer #1 (Remarks to the Author):

The authors has answer all of my suggestions, and from my perspective it is ready for publication.

Reviewer #2 (Remarks to the Author):

I think authors appropriately responde all question raised by reviewers